# CONTINUOUS SPEECH SYNTHESIS USING PER-TOKEN LATENT DIFFUSION

## ABSTRACT

The success of autoregressive transformer models with discrete tokens has inspired quantization-based approaches for continuous modalities, though these often limit reconstruction quality. We therefore introduce SALAD, a per-token latent diffusion model for zero-shot text-to-speech, that operates on continuous representations. SALAD builds upon the recently proposed expressive diffusion head for image generation, and extends it to generate variable-length outputs. Our approach utilizes semantic tokens for providing contextual information and determining the stopping condition. We suggest three continuous variants for our method, extending popular discrete speech synthesis techniques. Additionally, we implement discrete baselines for each variant and conduct a comparative analysis of discrete versus continuous speech modeling techniques. Our results demonstrate that both continuous and discrete approaches are highly competent, and that SALAD achieves a superior intelligibility score while obtaining speech quality and speaker similarity on par with the ground-truth audio.

## 1 INTRODUCTION

Autoregressive (AR) modeling is often correlated with discrete representations, probably due to the remarkable success of Large Language Models (LLMs), which operate on a discrete modality. (Vaswani et al., 2017; Radford et al., 2019). Inspired by the success of LLMs, continuous modalities, such as audio and images, are quantized to be modeled discretely. In image generation, quantization is often achieved by discrete autoencoders (Van Den Oord et al., 2017), which are later optimized with adversarial losses to improve fidelity (Esser et al., 2021). Works that focus on audio generation usually employ *Residual Vector Quantization* (RVQ) (Zeghidour et al., 2021), a process that iteratively refines the approximation by quantizing the residual. The resulting discrete codes are used for discrete AR modeling (Esser et al., 2021; Wang et al., 2023; Copet et al., 2024).

Discrete modeling over continuous domains requires quantization, which degrades the reconstruction quality and *upper-bounds* the fidelity. Using multiple RVQ quantizers enhances the fidelity, but the fine RVQ codes might *quantize noise*, which can be detrimental for discrete modeling methods. Discrete autoencoders may also suffer from low codebook utilization (Mentzer et al., 2023), and multimodal models that work on discrete representation suffer from stability issues (Team, 2024). We therefore suspect that quantizing inherently continuous modalities may be *sub-optimal*, and focus on continuous alternatives instead.

Predicting continuous distributions with regression losses such as L1 or L2, induce a unimodal distribution, an unrealistic assumption for most generative tasks. We hypothesize that multimodal distributions, which enable multiple local maxima, can represent more complex patterns and is crucial for generative one-to-many tasks. Recent works in image generation have explored approaches to modeling continuous distributions. GIVT (Tschannen et al., 2023) represents the continuous distribution using a Gaussian Mixture Model, while AR-Diffusion (Li et al., 2024) suggests a per-token diffusion head to model the continuous frame distributions.

We suggest SALAD (Speech synthesis with Autoregressive LAtent Diffusion), a per-token latent diffusion model for zero-shot speech synthesis over continuous representations, inspired by the per-token diffusion head suggested by Li et al. (2024). We enable the generation of *variable length* outputs, addressing a challenge that is absent in image generation methods, where the number of tokens to generate is fixed. We utilize semantic tokens (Kharitonov et al., 2023; Borsos et al., 2023a)

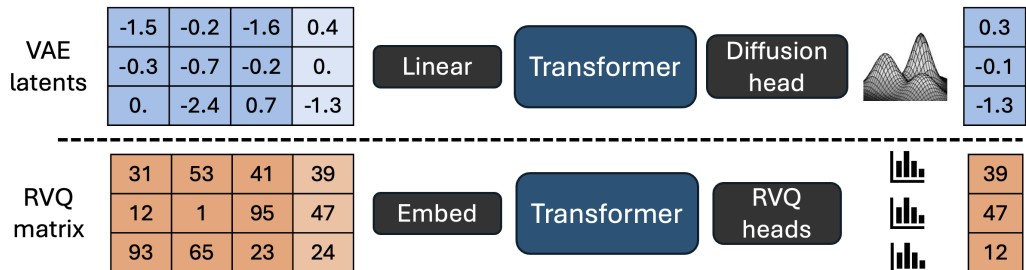

Figure 1: Continuous vs. discrete modeling

– quantized embeddings of a self-supervised model – for contextual information and to define the generation-stopping condition. SALAD does not rely on text-audio alignment, which makes it easier to leverage large data sources, and can synthesize audio based on a target speaker using a three-second speaker prompt. We propose three variants for SALAD:

1. *T2A* (Text2Acoustic): predicts acoustic features directly from text, using semantic tokens as an auxilary task.

2. *S2A-AR* (Semantic2Acoustic Autoregressive): predicts acoustic features from semantic tokens by next-token prediction.

3. *S2A-NAR* (Semantic2Acoustic Non-Autoregressive): predicts acoustic features from semantic tokens using a MaskGIT (Chang et al., 2022) schedule.

For each of our continuous variants, we train a comparable model that operates on *discrete* representations, replacing the diffusion head with RVQ discrete prediction heads (Figure 1). Our discrete T2A model is the first to predict semantic and acoustic tokens in parallel directly from text. Our SoundStorm discrete baseline employs the random unmasking method, which we demonstrate to outperform the confidence-based unmasking. We evaluate all models on speech quality, intelligibility, and speaker similarity. The results suggest that SALAD's T2A model achieves the highest intelligibility score, while having speech quality and similarity scores on-par with the ground-truth audio, as measured in subjective listening tests [1]. Our contributions can be summarized as follows:

• Propose SALAD, a zero-shot speech synthesis system that uses per-token latent diffusion.

• Extend popular discrete speech synthesis methods to continuous representations.

• Suggest a discrete text-to-acoustic model and improve SoundStorm's unmasking method.

• Compare discrete and continuous modeling techniques in a controlled environment.

## 2 RELATED WORK

**Zero-Shot TTS**  Inspired by the success of in-context learning, there has been great interest in text-to-speech (TTS) systems that can generalize to unseen speakers during inference. This task is commonly known as zero-shot TTS, and provides many benefits due to its flexibility and increased quality (Wang et al., 2023). Zero-shot TTS systems, including SALAD, typically formulate the problem as a language modeling task, containing text and audio tokens. Such methods make use of a speaker prompt – a short recording from the target speaker – and synthesize the text according to the prompt (Le et al., 2024; Shen et al., 2023; Łajszczak et al., 2024; Peng et al., 2024).

**Semantic Tokens**  Quantized embeddings of self-supervised audio models (Hsu et al., 2021; Baevski et al., 2020; Chung et al., 2021), commonly known as semantic tokens, have been shown to capture phonetic and prosodic information, and show benefits when modeling long-range dependencies in various speech processing tasks. These tokens became popular as intermediate represen-

---

[1]Samples are available at https://s3.us-south.objectstorage.softlayer.net/zk-wav-data/Webpages/ICLR2025PerTokenLatentDiffusion/index.html

tations for speech synthesis (Kharitonov et al., 2023; Huang et al., 2023; Borsos et al., 2023a), unconditional audio generation (Borsos et al., 2023b), and for text-audio multimodal tasks (Rubenstein et al., 2023). SALAD predicts semantic tokens as an auxilary task to obtain contextual information and determine the stopping condition.

**RVQ codes prediction** Various audio generation methods have designed unique ways to predict the RVQ codes matrix, each with their advantages and limitations. AudioLM (Borsos et al., 2023b) flattens the codes matrix into a long sequence, which greatly increases the token count in transformer models. Most followup works avoid flattening and embed all RVQ residual layers into a single token. Vall-E (Wang et al., 2023) generates the initial coarse code vector with an AR model and then uses an non-autoregressive (NAR) model to predict the rest of the codebooks. AudioGen (Kreuk et al., 2022) uses a single AR model to predict all codes of each timestep in parallel. MusicGen (Copet et al., 2024) extends AudioGen by introducing a delay pattern, ensuring each code is predicted based on its coarser RVQ layers, leading to a better approximate factorization. SoundStorm (Borsos et al., 2023a) employs the fast NAR decoding algorithm by MaskGIT (Chang et al., 2022) to generate acoustic tokens based on semantic tokens. NaturalSpeech3 (Ju et al., 2024) trains a factorized codec, which disentangles speech characteristics into discrete factors and predicts each factor using a MaskGIT procedure. As opposed to all above methods, SALAD directly predicts a continuous latent space, thus avoiding the need to predict multiple residuals codes.

**Continuous models** When learning a continuous distribution, recent works typically use diffusion models, which were developed to sample from complex continuous probability distributions, inspired by non-equilibrium thermodynamics (Ho et al., 2020). Several works attempt to synthesize speech using a diffusion process, which has the challenge of generating variable length outputs (Kong et al., 2020; Chen et al., 2020; Popov et al., 2021). For that end, most diffusion-based works rely on a duration predictor that predicts the audio length in advance, which might be inferior to determining the length on-the-fly during synthesis (Shen et al., 2023; Le et al., 2024). MELLE (Meng et al., 2024) predicts Mel spectrograms autoregressively using a Gaussian sampling module, and parameterizes the next frame using a Gaussian distribution, which restricts it to learn only unimodal distributions. MELLE relies on an additional binary classifier that indicates when to stop, which is a highly imbalanced classification problem. In contrast, SALAD operates on VAE latent tokens, which allows sampling diverse inputs while training, and uses a diffusion head, capable of modeling multimodal distributions. SALAD relies on semantic-tokens to determine the stopping condition, a more balanced representation which also provides contextual information.

## 3 METHOD

### 3.1 BACKGROUND

**Definitions** We denote the raw audio sequence as $\boldsymbol{a} = (a_1, ..., a_m)$ where $a_i \in [-1, 1]$ with sampling rate $f_S$. The text is $\boldsymbol{y} = (y_1, .., y_k)$ where $y_i \in \mathcal{A}$, and $\mathcal{A}$ is the text vocabulary. We obtain compressed audio representations using a variational autoencoder (VAE), trained with adversarial losses to obtain high-fidelity reconstructions. The VAE's encoder $\mathcal{E}$ predicts a sequence of means and variances of normal distribution: $(\mu_1, ..., \mu_n), (\sigma_1^2, ..., \sigma_n^2) = \mathcal{E}(\boldsymbol{a})$ where $\sigma_i, \mu_i \in \mathbb{R}^d$ and $d$ is the VAE bottleneck dimension. The VAE downsamples the sequence with a stride $r$. We sample $x_i \sim \mathcal{N}(\mu_i, \sigma_i^2)$ and denote $\boldsymbol{x} = (x_1, .., x_n)$ as the continuous *acoustic tokens*. The VAE's decoder $\mathcal{D}$ is used for reconstruction $\hat{a}_1, ..., \hat{a}_m = \mathcal{D}(x_1, .., x_n)$. We also extract semantic tokens and denote them by $\boldsymbol{w} = (w_1, .., w_m)$, which have the same downsampling stride as the VAE. Our goal is to predict the audio based on the desired text and the speaker prompt. Denoting the speaker prompt latent features as $\boldsymbol{s} = s_1, ..., s_p$, our training objective can be formulated by: $p(\boldsymbol{x}|\boldsymbol{y}, \boldsymbol{s})$.

**Diffusion Process** A diffusion process starts from a continuous signal, and gradually destroys it using a forward noise process. Our method performs latent diffusion, and attempts to predict the VAE latent vectors $x_1, ..., x_n$. Given noising coefficients $\beta_0, ..., \beta_T$ and some continuous vector $x$, we define $x^0 = x$ and $\epsilon \sim \mathcal{N}(0, I)$; the Markov structure is $x^t = \sqrt{1 - \beta_t}x^{t-1} + \sqrt{\beta_t}\epsilon$. This iterative denoising process can be simplified. By defining $\alpha_t = 1 - \beta_t$ and $\bar{\alpha}_t = \prod_{i=1}^{t} \alpha_i$, we get that $x^t = \sqrt{\bar{\alpha}_t}x + \sqrt{1 - \bar{\alpha}_t}\epsilon$. The diffusion process is often defined such that $\bar{\alpha}_T \to 0$ and $x^T$ distributes closely to the standard normal distribution. Diffusion models $\epsilon_\theta$ are trained to perform

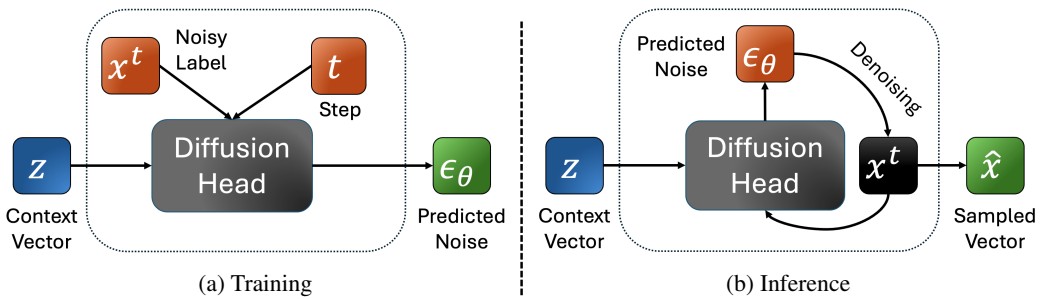

Figure 2: The per-token diffusion head

the reverse diffusion process, which denoises the corrupted signal by predicting the added noise. Their denoising loss is defined as $\mathcal{L}(x) = \mathbb{E}_{\epsilon,t}\left[\|\epsilon - \epsilon_\theta(t, x^t)\|^2\right]$. Most diffusion models operate on a sequence $x_1, .., x_n$ and attempt to denoise all tokens in parallel using $\epsilon_\theta(t, x_1^t, ..., x_n^t)$.

**Per-Token Diffusion Head** Li et al. (2024) proposed an MLP diffusion head for image generation. Unlike standard diffusion models, the diffusion head denoises each token *independently*, which gives additional flexibility when defining the conditioning information (e.g., predicting on previously predicted tokens). We rely on a transformer model $\Theta$ that extracts contextual per-token conditioning vectors $z_1, .., z_n$ based on the input features and optional context vectors that we denote by $C$

$$\boldsymbol{z} = z_1, ..., z_n = \Theta(C, x_1, ..., x_n)$$

The diffusion head (noise estimator) $\epsilon_\theta$ takes a contextual conditioning vector $z$ and attempts to model the continuous distribution $p(x|z)$. Given a target token $x$, we follow a similar diffusion process but condition the prediction on $z$. The loss is

$$\mathcal{L}(x, z) = \mathbb{E}_{\epsilon,t}\left[\|\epsilon - \epsilon_\theta(x^t, t, z)\|^2\right] \quad (1)$$

During training, we sample $t \sim [T], \epsilon \sim \mathcal{N}(0, I)$ for each token $x$, obtain the noisy targets $x^t$, and minimize $\mathcal{L}(x, z)$ (Figure 2a). This denoising network is trained jointly with the transformer $\Theta$, and the gradient with respect to $z$ is propagated to the transformer. We can sample $K$ different values of $t, \epsilon$ for a given context vector and target $z, x$, with the additional complexity of just the MLP head rather than the entire model. During inference, we sample a continuous vector by sampling a Gaussian vector $x^T \sim \mathcal{N}(0, I)$ and reversing the diffusion process (see Figure 2b):

$$x^{t-1} = \frac{1}{\sqrt{\alpha_t}}\left(x^t - \frac{\beta_t}{\sqrt{1 - \bar{\alpha}_t}}\epsilon_\theta(x^t, t, z)\right) + \sqrt{\beta_t}\epsilon \quad (2)$$

## 3.2 SALAD: Speech Synthesis using Autoregressive LAtent Diffusion

SALAD performs zero-shot text to speech, which can synthesize speech based on a given text and a speaker prompt. It does so by predicting the continuous VAE latents $\boldsymbol{x}$ with the per-token diffusion head. Our approach utilizes semantic tokens $\boldsymbol{w}$ as an auxiliary representation that provides contextual information and determines the stopping condition. We provide two variants for SALAD:

- Semantic to Acoustic (S2A) - predicts acoustic features based on semantic tokens, and relies on an external text-to-semantic model to produce the semantic tokens (Figure 3).
- Text to Acoustic (T2A) - predicts acoustic features and semantic features directly from text, relying on the stopping condition of the semantic tokens (Figure 4).

### 3.2.1 Semantic to Acoustic (S2A)

Following Kharitonov et al. (2023), we divide synthesis into two tasks: text-to-semantic (T2S) and semantic-to-acoustic (S2A), each tackled by a different model. The T2S model predicts the discrete semantic tokens based on the text and speaker autoregressively using a causal transformer:

$$p(w_1, .., w_n | t, s) = \prod_{i=1}^{n} p(w_i | t, s, w_1, .., w_{i-1})$$

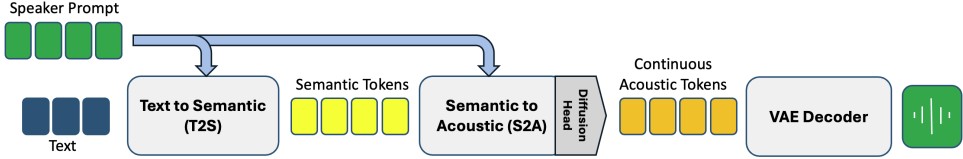

Figure 3: Synthesis using Semantic-to-Acoustic models

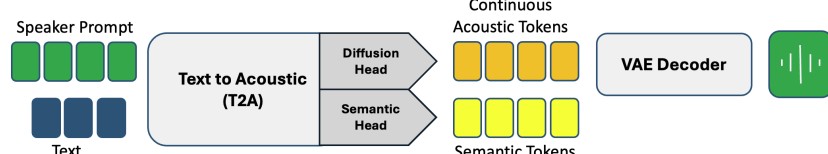

Figure 4: Synthesis using Text-to-Acoustic models

Importantly, the T2S model determines the synthesized audio length. Then, our S2A model predicts the acoustic tokens based on the semantic tokens, using an additional transformer model with the diffusion head to model the continuous distributions. We provide two variants of semantic-to-acoustic models, AR and NAR, following the literature on discrete acoustic token modeling.

**Autoregressive S2A** In our AR semantic-to-acoustic model, our training objective is $p(x_1, ..., x_n | \boldsymbol{w}, \boldsymbol{s}) = \prod_{i=1}^{n} p(x_i | \boldsymbol{w}, \boldsymbol{s}, x_1, .., x_{i-1})$. We input the latent frames, the semantic tokens, and speaker prompt into a causal transformer and obtain the contextual condition vectors $z_1, .., z_n = \Theta(\boldsymbol{w}, \boldsymbol{s}, x_1, ..., x_n)$. The frame $x_i$ is predicted given $z_{i-1}$ and our loss is:

$$\mathcal{L}(\boldsymbol{x}, \boldsymbol{z}) = \sum_{i=1}^{n} \mathbb{E}_{\epsilon_i, t_i} \left[ \|\epsilon_i - \epsilon_\theta(x_i^t, t_i, z_{i-1})\|^2 \right]$$

During inference, the T2S model generates semantic tokens $\hat{\boldsymbol{w}}$. Then, the S2A model generates the continuous latent vectors based on the predicted semantic tokens, by computing the contextual embedding $z_i$, and uses diffusion head inference to sample the next continuous frame $x_{i+1}$.

**Non-Autoregressive S2A** MaskGIT (Chang et al., 2022) trains a bidirectional transformer on a discrete masked language modeling objective. During inference, it unmasks tokens over $K$ inference steps, where each step is based on the previously predicted tokens. Soundstorm extended this procedure to predict the RVQ codes based on semantic tokens, by applying the MaskGIT procedure for each RVQ layer. We extend the MaskGIT procedure to predict continuous acoustic tokens based on semantic tokens, using the diffusion head defined in Section 3.1. Given a schedule function $\gamma(r) : [0, 1] \rightarrow [0, 1]$ and a sequence $x_1, .., x_n$, we sample $r \in U[0, 1]$ and mask out $\gamma(r) \cdot n$ acoustic tokens. Define the random masking indicators $\boldsymbol{m} = (m_1, ..., m_n)$ where $m_i \in \{0, 1\}$, we replace masked acoustic tokens with a fixed learnable embedding $q$ and define $r_i = m_i \cdot q + (1 - m_i) \cdot x_i$. As in SoundStorm, a semantic token $w_i$ and masked acoustic tokens $r_i$ are embedded into a single transformer token, and speaker prompt tokens are left unmasked and appended as context. The resulting sequence is fed into a bidirectional transformer, and the resulting contextual vectors $z_1, .., z_n$ are used to predict the masked acoustic tokens using the diffusion head. The loss function is the denoising of the masked acoustic tokens

$$\mathcal{L}(\boldsymbol{x}, \boldsymbol{z}, \boldsymbol{m}) = \sum_{i=1}^{n} \mathbb{E}_{\epsilon, t} \left[ m_i \cdot \|\epsilon - \epsilon_\theta(x_i^t, t, z_i)\|^2 \right]$$

During inference, MaskGIT chooses the tokens with the highest confidence. In a continuous diffusion model, measuring the confidence is not trivial, so we opt to select the tokens to unmask at random. We first check the influence of confidence-based selection in MaskGIT inference in Section A.2, and conclude that random unmasking is surprisingly *superior* to the confidence-based unmasking, which validates our design choice. Specifically, given a sequence of $n$ tokens, $K$ maskGIT steps, in every step $k \in [K]$ we unmask $n \left( \gamma \left( \frac{k}{K} \right) - \gamma \left( \frac{k-1}{K} \right) \right)$ additional tokens selected at random, by applying the per-token diffusion head.

### 3.2.2 Text-to-Acoustic (T2A)

Decoupling TTS into T2S and S2A requires training two models of similar size, and applying two steps of inference, which greatly increases the compute requirements and latency. Therefore, we suggest an end-to-end text-to-acoustic model (T2A) which predicts the acoustic features directly from the text and the speaker prompt. The T2A model predicts the semantic and acoustic features in parallel, where the semantic token prediction is an auxiliary task that allows conditioning on contextual information and provides a stopping condition. We add an additional prediction MLP to predict the discrete semantic tokens. We adopt the delay pattern suggested by (Copet et al., 2024) such that every acoustic token $x_i$ can be predicted based on the semantic token $w_i$. Define $r_i = (w_i, x_{i-1})$, we extract contextual features from our transformer backbone, based on the text and speaker prompt $z_i = \Theta(t, s, r_1, ...r_i)$, which is used to predict $w_{i+1}$ using the cross-entropy loss $L_s$ and $x_i$ using the diffusion loss $L_a$. We weigh the two losses to $\mathcal{L} = \alpha\mathcal{L}_a + (1 - \alpha)\mathcal{L}_s$. We halt the generation after the semantic prediction head samples an EOS token. We note that the audio duration is predicted on the-fly based on the model's predictions, unlike most diffusion-based TTS models, where the audio duration is predetermined.

### 3.3 Discrete Baselines

All SALAD models use common discrete architectures and only replace the input projection and prediction head, so we can implement a discrete variant for each method proposed in Section 3.2. The discrete methods use an RVQ-GAN quantizer, which yields a sequence of discrete codes $(q^1, ..., q^Q)$ for each frame. These codes are predicted from the contextual embedding $z$ by MLP prediction heads, one for each codebook. The S2A-AR discrete model predicts all codes in parallel, using the delay pattern proposed in MusicGen (Copet et al., 2024). The S2A-NAR discrete model implements SoundStorm (Borsos et al., 2023a), which applies $Q$ MaskGIT procedures, one for each codebook. Unlike the vanilla SoundStorm implementation, we replace the confidence-based unmasking with random unmasking, as we have found it to be superior. Given $K$ MaskGIT steps, SoundStorm requires $QK$ passes through the transformer, as it employs a MaskGIT procedure per RVQ layer, unlike SALAD-NAR, which requires $K$ transformer passes. The T2A discrete model predicts semantic and acoustic tokens in parallel from text, using the delay pattern described above. We use the MusicGen parallel prediction method, treating the semantic tokens as the coarsest codes. In our listening test, we compare to the external XTTS (Casanova et al., 2024), a commercial SOTA zero-shot TTS model.

## 4 Experiments

### 4.1 Experimental Setup

**Datasets**  We train all models on the English subset of multi-lingual LibriSpeech (MLS) (Pratap et al., 2020), which contains 10M examples of 10-20 seconds, resulting in 45K hours. To avoid over-exposure of a few speakers, we limit the maximal number of utterances per speaker to 10K, resulting in 5.2M examples. We evaluate all models on LibriSpeech *test-clean* (Panayotov et al., 2015), which consists of 2620 utterances by 40 speakers. All speakers in the test set are excluded from the training set. We filter the dataset to utterances with lengths of 8-25 seconds, and then limit to at most 15 samples per speaker, resulting in 564 utterances for evaluation.

**Tokenization**  To derive acoustic tokens, we train continuous $\beta$-VAE-GAN, with a varying bottleneck dimension $d \in \{8, 16, 24, 32\}$, and set the KL-divergence regularization to $\beta = 5 \cdot 10^{-5}$, as done in Tschannen et al. (2023). We also train discrete RVQ-GAN models with $q \in \{4, 8, 12\}$ codebooks, each with 1024 entries. In addition, we apply quantizer dropout (Zeghidour et al., 2021) with $p = 0.5$. All compression models are trained on MLS-English, DAPS, LibriTTS, LibriTTS-R and LJ-Speech, which balance between high and mid quality recordings (Shechtman & Dekel, 2024). The all-training hyperparameters follow the original recipe proposed by Kumar et al. (2024). We extract semantic tokens by quantizing the embeddings of the 11th layer of W2V-BERT (Barrault et al., 2023) using minibatch K-means with 1024 centroids. We further compress the semantic tokens using a BPE tokenizer with a vocabulary of 16384 tokens. This is done to shorten the sequence and balance the tokens' distribution (Dekel & Fernandez, 2024). We note that only the T2S model

leverages the BPE-compressed semantic tokens, as well as the the S2A-AR models, as other models embed semantic and acoustic features into a shared token space. We also train a text BPE tokenizer on the transcripts of our training set, with a vocabulary of 16384 tokens.

**Architecture**   We use a transformer backbone with $d = 1024$, $d_{ff} = 4096$, 24 layers, 16 heads, sinusoidal positional embedding, GeLU activation, and a dropout rate of $0.1$, resulting in models with roughly 350M parameters. VAE embeddings are projected using a linear layer, while RVQ tokens are embedded using $Q$ lookup tables, which are summed into a single embedding. We use Classifier-Free Guidance (Ho & Salimans, 2022) and randomly omit the speaker prompt with $p = 0.1$ during training. In the MaskGIT NAR experiments, we use the cosine masking schedule, and apply a total of 64 inference steps, where the SoundStorm model with 4 codebooks performs 16 inference steps per layer. RVQ codes are predicted using a $Q$ MLP heads with four hidden layers. We sample from discrete distributions using Top $k = 10$ sampling, with a temperature of $\tau = 1$, a repetition penalty of $1.05$, and a CFG scale of $\alpha = 3$.

We use a diffusion process with $T = 1000$ steps, where betas are logarithmicly spaced between $\beta_0 = 2e - 4$ and $\beta_T = 0.03$. Our per-token diffusion head is an MLP network with 12 residual layers, that predicts the noise $\epsilon$ given the transformer embedding vector $z$, the noisy input $x^t$, and the diffusion step $t$. Each residual block consists of layer normalization, linear layer, SiLU activation, and dropout with $p = 0.1$. During inference, we apply 20 diffusion steps for sampling, with a default noise scale of 1. We use the AdamW optimizer, with $lr_{max} = 3e - 4$ and $lr_{min} = 3e - 5$, weight decay $0.1$, and a clip gradient norm of 1, and train with FP16 mixed precision. We linearly warm up the learning rate from $lr_{min}$ across 32K iterations to $lr_{max}$ and decay the learning rate back to $lr_{min}$ over 300K steps using a cosine schedule. Each global batch size has approximately 150K acoustic tokens (200 samples). Each model was trained with 8 A100 80GB GPUs.

**Metrics**   We measure *Audio Quality* using UTMOS (Saeki et al., 2022) which produces a quality score in the range of 1-5 (higher is better). *Intelligibility* is measured by the character error rate (CER) in percentages (%) between the ground-truth text and the Whisper transcripts (Radford et al., 2023) of the synthesized audio. *Speaker Similarity* is measured by the cosine similarity to the prompt, comparing the embedding of WavLM-TDNN (Chen et al., 2022), a popular speaker verification model. This metric was also reported in Vall-E and subsequent studies (Wang et al., 2023; Chen et al., 2024). The similarity score predicted is in the range of $[-1, 1]$, where a larger value indicates a higher similarity.

For the subjective *Listening Tests*, we selected one random utterance for every speaker in LibriSpeech *test-clean* (20 female and 20 male speakers), resulting in 40 utterances for evaluation. The selected utterances were confined to have at most 200 characters to enable the comparison with XTTS *xttsv2* (Casanova et al., 2024) (*xttsv2* demo limitation). For each sample, we selected a three-second-long speaker prompt from another random utterance of the same speaker. Each system synthesizes the desired utterance based on the same text and speaker prompt. All experiments were conducted on the Amazon Mechanical Turk (AMT) crowd-sourcing platform with votes collected from 39-58 subjects qualified as *masters* (Sodré & Brasileiro, 2017).

In the first Listening Test we assess speech quality and naturalness by the standard 5-point scale Mean Opinion Score (MOS) (Ribeiro et al., 2011). 25 distinct subjects assessed each utterance. We report the average scores and the 95% confidence interval. In the second Listening Test we asses the Speaker Similarity by a 4-level pairwise similarity test, as in (Wester et al., 2016; Kons et al., 2018), where subjects were presented with *(utterance, prompt)* pairs and asked to rank speaker similarity of each pair on a 4-level categorical scale *(definitely different speakers, probably different speakers, probably the same speaker, definitely the same speaker)*. Each utterance was assessed by 20 distinct subjects on average. We report the mean similarity score and the 95% confidence interval while attaching 1-4 numerical values to the above categories, as in (Kons et al., 2018).

### 4.2   RESULTS

**Objective Evaluation**   We evaluate all models on zero-shot TTS. Given a text and a three-second speaker prompt, which is taken randomly from another utterance of the same speaker, the model attempts to synthesize the audio with the identity and prosody similar to the prompt. All models use the same random prompt for each sample. We compare two variants of models that perform Se-

| Task | Modeling | Representation | UTMOS ↑ | STT CER (%) ↓ | Similarity ↑ |
|------|----------|---------------|---------|---------------|--------------|
| Ground Truth | – | – | 4.121 | 0.528 | 0.736 |
| Text to Acoustic | AR | Continuous | **4.280** | **0.739** | 0.539 |
| Text to Acoustic | AR | Discrete | 4.270 | 2.298 | **0.600** |
| Semantic to Acoustic | AR | Continuous | 4.27 | 2.198 | **0.588** |
| Semantic to Acoustic | AR | Discrete | **4.348** | **1.231** | 0.549 |
| Semantic to Acoustic | NAR | Continuous | 4.277 | **1.393** | 0.558 |
| Semantic to Acoustic | NAR | Discrete | **4.351** | 1.846 | **0.602** |

Table 1: Objective evaluation of LibriSpeech *test-clean*

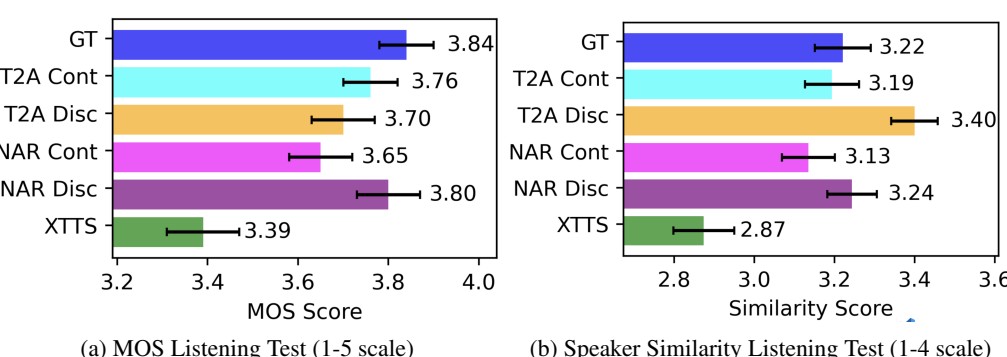

(a) MOS Listening Test (1-5 scale)  (b) Speaker Similarity Listening Test (1-4 scale)

Figure 5: Subjective listening results

mantic to Acoustic (S2A), and another variant that performs Text to Acoustic (T2A) directly. When using S2A models, we first run the *Text to Semantic* (T2S) model and use the predicted semantic tokens, as in Figure 3. The discrete models rely on a 4 codebook model, while the continuous make use of a $d = 8$ VAE embedding. Table 1 shows that the continuous models are competitive with their discrete benchmarks. The continuous T2A model presents the highest intelligibility score, making it the most reliable model when having to synthesize an exact text. However, the speaker similarity scores of the discrete T2A and S2A-NAR model are higher. We note that in cases of accented speech or low quality recordings, when the speaker similarity increases, the intelligibility and audio quality often decreases. We did not report objective scores for XTTS due to the sample limit in their demo.

**Subjective Evaluation**   We conduct the two subjective listening tests, described above, to compare the following systems: (1) Ground Truth audio (2) XTTSv2 (Casanova et al., 2024) (3) T2A Continuous (4) T2A Discrete (5) S2A NAR Continuous (6) S2A NAR Discrete. Figure 5a reports the mean opinion score (MOS) results, suggesting that the difference between the ground-truth audio (GT) to both T2A continuous model and the NAR discrete model is statistically insignificant ($p > 0.01$). Figure 5b presents the speaker similarity average score with 95% confidence intervals, suggesting similar or better speaker similarity scores for all the systems but *XTTSv2*. More precise analysis with two-sided Wilkinson rank-sum test (Wilcoxon, 1945) reveals that both the T2A continuous and the NAR discrete models do not differ ($p >> 0.01$) from the GT in terms of speaker similarity, while the T2A discrete model is marginally better than the GT ($p = 0.0105$). The NAR continuous model, however, is marginally worse than the GT ($p = 0.0111$).

## 4.3   ABLATION STUDY

**Inference Hyperparameters**   We now turn to investigate the influence of inference hyperparameters on synthesized speech. We used the T2A model to investigate classifier-free guidance (CFG), noise scale and the number diffusion steps, and the S2A-NAR model for the MaskGIT inference experiment. In every experiment, we fix all values to the default inference values following those described in Section 4, and change only a single hyperparameter. The CFG linear extrapolation coefficient increases the speaker similarity, but degrades the automatic quality metric, as seen in

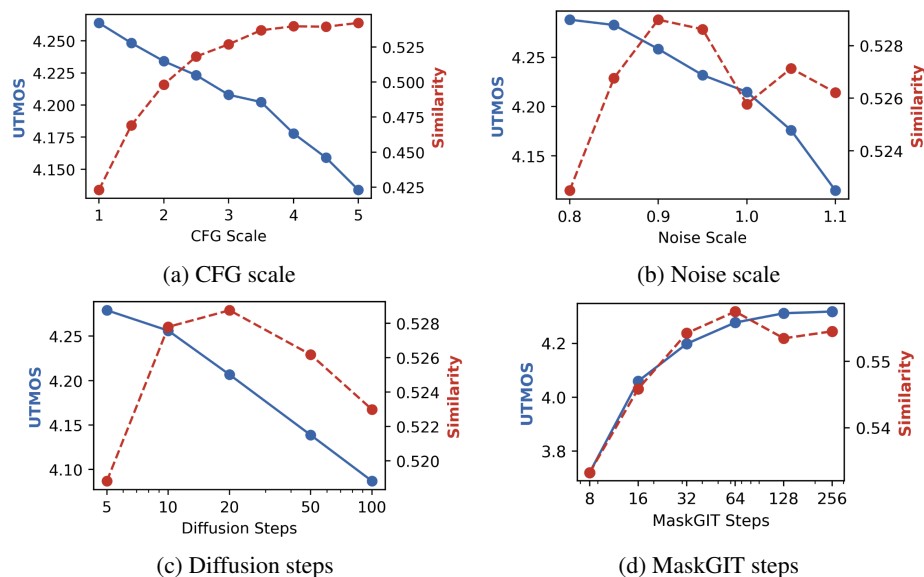

Figure 6: Inference hyperparameters influence

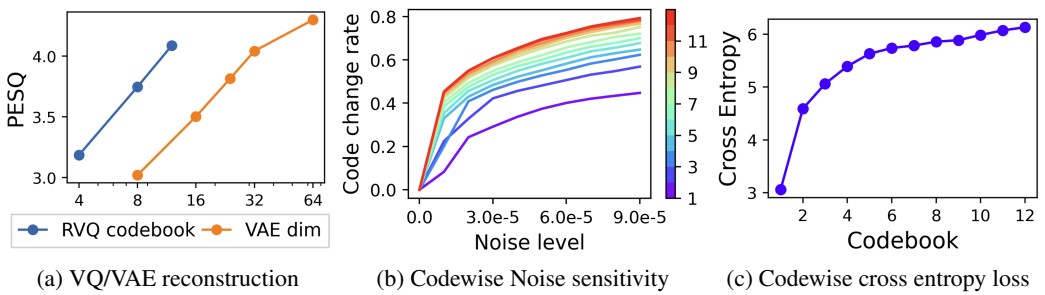

Figure 7: High-fidelity RVQ codes

Figure 6a. Next, we scale the noise level added in each diffusion step by scaling the $\beta_t \epsilon$ term in Equation 2, and see improvements in similarity but degradation in the UTMOS quality score (Figure 6b). We also examine the number of diffusion steps, which improve similarity until reaching 20 diffusion steps, and also degrade UTMOS (Figure 6c). The number of MaskGIT in the NAR model shows consistent improvement in both the speaker similarity and UTMOS (Figure 6d).

**High-Fidelity Modeling** When increasing the number of RVQ codebooks or the VAE embedding dimension, the reconstruction quality increases, but language modeling can be difficult (Shen et al., 2023). Figure 7a shows the reconstruction quality measured by PESQ (Rix et al., 2001), for different numbers of RVQ codebooks and VAE embedding dimensions. One concern regarding RVQ modeling is that the fine codes quantize noise, leading to a high gradient contribution of random classification problems. We measure the noise sensitivity per codebook by adding Gaussian noise into raw samples, compressing them with the RVQ model, and checking the ratio of change per codebook. Results in Figure 7b suggest that fine codebooks are indeed more sensitive to noise. In Figure 7c, we calculate per-codebook validation cross-entropy loss in the discrete 12-codebook T2A-AR model, suggesting the model struggles to reduce uncertainty in finer codebooks. This phenomenon occurs despite the delay-pattern described in Section 3, where the finer RVQ levels are conditioned on coarser layers of the same frame. Finally, we compare the generation quality with less-compressed representations. The results in Table 2 show that when increasing the fidelity, the intelligibility drop of the high-fidelity discrete T2A is greater comparing to the continuous model.

|  | UTMOS ↑ | Intelligibility ↓ | Similarity ↑ |
|---|---|---|---|
| T2A HiFi Continuous $d = 32$ | **4.271** | **1.157** | 0.545 |
| T2A HiFi Discrete $Q = 12$ | 4.203 | 5.461 | **0.597** |

Table 2: Discrete vs continuous models with high-fidelity representations

|  | UTMOS ↑ | Intelligibility ↓ | Similarity ↑ |
|---|---|---|---|
| VAE Sample | **4.280** | **0.739** | 0.539 |
| VAE NoSample | 3.468 | 1.891 | **0.613** |

Table 3: Influence of VAE sampling during training

**VAE sampling** VAE models allow the sampling of diverse inputs, unlike the discrete codebooks or Mel spectrograms. This ability may improve the robustness of the model, and better account for the mismatch between training and inference (during inference, the model predictions are based on its previous noisy predictions). To check the influence of VAE sampling, we compare two T2A models - one samples from the VAE distribution $x = \mu + \epsilon \cdot \sigma$ and the other always takes the mean $x = \mu$. The results in Table 3 show a large gap in UTMOS and intelligibility indicating that sampling improves synthesized samples. We listened to audio samples from the VAE-NoSample model, and noticed a gradual addition of speaker-inconsistency artifacts throughout the synthesis. We suspect that the addition of VAE-sampling noise during training made it more robust to the mismatch between training and inference. We also hypothesize that high similarity result of VAE-NoSample is caused by the artifacts described above.

## 5 DISCUSSION

Compressing complex signals such as audio and images often introduces a tradeoff between *perception* and *generation*. For tasks involving perception or understanding, compression can lead to information loss, resulting in degraded performance. However, for generation, compression has proven to be highly effective, as the generative model has to learn a lower-dimensional distribution. Multimodal models typically aspire to work with symmetric representations, in which the input and output representations are identical, as commonly done in language models. Developing generative methods that can operate upon less-compressed representations would alleviate the perception-generation models, and improve multimodal models that operate on symmetric representations. While RVQ is a powerful compression mechanism, capable of providing high-fidelity representations, it may lead to noise quantization. Working with continuous representations can be more robust to noise, as continuous models scale the noise according to its magnitude.

**Limitations** The diffusion head inference process is slower than the RVQ prediction heads, as it requires an iterative process for the token sampling. Moreover, it does not allow to measure likelihood or confidence, which can be useful for decoding algorithms such as beam search or confidence based unmasking. Optimal balancing of the discrete and continuous losses in the continuous T2A model is not easy to obtain. During training, the gradients of the discrete semantic loss increase, while the gradients of the continuous diffusion loss decrease.

**Future work** Follow-up works can extend our work and develop multimodal models that operate on symmetric representations, and are capable of perception and generation. They can also derive a generation stopping condition that does not rely on any discrete representation. Additionally, future works can implement diverse inference strategies that adapt the number of diffusion steps per token (e.g. more diffusion steps in the first tokens), or develop a quality metric for a diffusion process, to allow decoding algorithms such as beam-search to be used during inference.

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

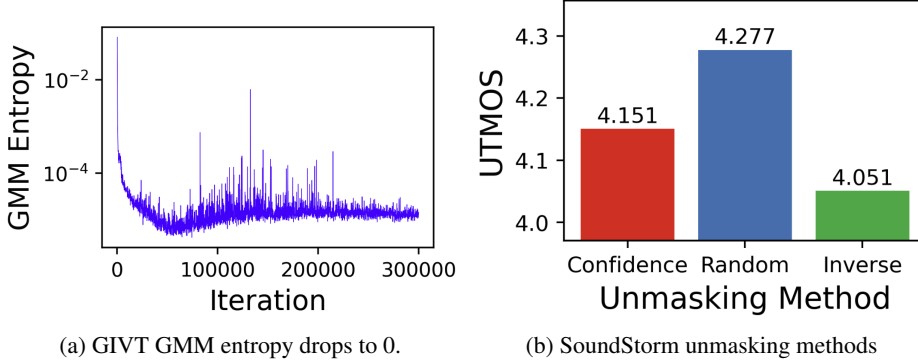

(a) GIVT GMM entropy drops to 0.    (b) SoundStorm unmasking methods

Figure 8: Additional results

## A    ADDITIONAL RESULTS

### A.1    GIVT

We also attempted to use GIVT (Tschannen et al., 2023) as an alternative approach for continuous audio generation. GIVT models the next token distribution as a Gaussian mixture model (GMM). We trained a GIVT model to a mixture of 16 Gaussians, which predicts the next continuous acoustic frame. We focused on the ability to produce multi-mode distributions. Figure 8a plots the entropy of mixture coefficients in the GMM, which drops quickly to zero during training. This might suggest that the ability to produce multimodal probability distributions is not being leveraged frequently.

### A.2    MASKGIT INFERENCE

In S2A-NAR-Cont, the MaskGIT selection of tokens to unmask is done at random, instead of being based on confidence scores. To check the influence of the unmasking approach, we provide three unmasking criteria for our SoundStorm model: highest confidence, random, and lowest confidence. The test was based on GT semantic tokens (to avoid dependency on semantic token prediction), with the default hyperparameters described in Section 4. We first sampled each token using top-k sampling, and defined the token score to be the softmax probability of the sampled token. We then unmasked tokens based on the score, its inverse, or at random. The results in Figure 8b suggest that random selection in SoundStorm yields improved performance over confidence-based selection. This resembles the results seen when using greedy sampling, which leads to a sub-optimal result.

