# OpenReview forum: "Continuous Speech Synthesis using per-token Latent Diffusion"
_ICLR.cc/2025/Conference — ICLR 2025 Conference Withdrawn Submission_

### Official Review · Reviewer_tscL · 2024-10-28

**Soundness:** 2
**Presentation:** 3
**Contribution:** 3
**Rating:** 6
**Confidence:** 4

**Summary:**

This paper proposed SALAD, a per-token latent diffusion model for zero-shot text-to-speech, that operates on continuous representations. Three variants of SALAD, including T2A (Text2Acoustic), S2A-AR (Semantic2Acoustic Autoregressive) and S2A-NAR (Semantic2Acoustic Non-Autoregressive) are explored. In addition, discrete and continuous modeling techniques are compared in a controlled environment.

**Strengths:**

1. Although the per-token diffusion head is very similar to [1], this paper is the first work applying such methods on speech and audio synthesis field. I believe the contributions of this paper is valuable to the community. In addition, the one-stage end-to-end Text-to-Acoustic model is interesting.
2. The experimental setup and comparisons are adequate and comprehensive, including various framework (two-stage S2A or one-stage T2A), paradigm of generating (autoregressive or non-autoregressive), and representations (discrete or continuous).
3. The paper is well-structured and written.


[1]. Tianhong Li, Yonglong Tian, He Li, Mingyang Deng, and Kaiming He. Autoregressive image generation without vector quantization. arXiv preprint arXiv:2406.11838, 2024.

**Weaknesses:**

1. The trends observed from experimental results are not clear. The Abstract states "both continuous and discrete approaches are highly competent". In addition, for various (tasks, modeling) combinations in Table 1, both discrete and continuous features have their merits. Therefore, it seems a bit unreasonable to use this title in this context, "**CONTINUOUS** SPEECH SYNTHESIS USING PER-TOKEN LATENT DIFFUSION".
2. There is no comparison of the proposed SALAD model with other open-source models. Such comparisons make the paper more convincing.
3. There is no comparison of the proposed SALAD model with other baselines without pre-token latent diffusion methods, which is very important for verifying motivation.
4. Time consumption is more important for sequence generation tasks, such as speech generation in this paper than image generation. The authors should give the RTF or other metrics on inference time in Table 1 as a reference.
5. I don't think the ablation study of VAE sampling makes much sense, and the authors should include more experiments with "Discrete Q"  or "Continuous d" to further illustrate the relationship between them and the quality of generation.

**Questions:**

1. For "We further compress the semantic tokens using a BPE tokenizer with a vocabulary of 16384 tokens.", can the author elaborate on this? How to compress 1024 categories of semantic tokens into 16384 categories?
2. "We also examine the number of diffusion steps, which improve similarity until reaching 20 diffusion steps, and also degrade UTMOS (Figure 6c).". Why more diffusion steps lead to worse performance? The authors should give a little explanation and speculation based on the observed phenomena.

---

### Official Review · Reviewer_XULT · 2024-10-31

**Soundness:** 2
**Presentation:** 3
**Contribution:** 2
**Rating:** 3
**Confidence:** 5

**Summary:**

The paper introduces SALAD, a per-token latent diffusion model for zero-shot text-to-speech synthesis that operates on continuous representations. SALAD extends the expressive diffusion head for image generation to generate variable-length outputs for speech synthesis. It utilizes semantic tokens for contextual information and determining the stopping condition. The authors propose three continuous variants of SALAD, extending popular discrete speech synthesis techniques, and implement discrete baselines for comparison. The results show that SALAD achieves superior intelligibility scores while maintaining speech quality and speaker similarity comparable to the ground-truth audio.

**Strengths:**

1. The paper presents a novel per-token latent diffusion model for continuous speech synthesis, which is a significant departure from traditional discrete modeling techniques.
2. Semantic Token Utilization: The use of semantic tokens for contextual information and generation-stopping conditions is a thoughtful integration that adds depth to the model's capabilities.
3. The paper includes a comparative analysis between continuous and discrete speech modeling techniques, which provides valuable insights into the performance of each approach.

**Weaknesses:**

Limited Comparison with Prior Work:

The paper's comparison with previous work is insufficient. It does not thoroughly engage with the existing body of literature on speech synthesis, particularly in terms of how SALAD's performance compares to state-of-the-art models on various metrics, such as the vall-E series including VALLE 2, RALLE and the naturalspeech series. The author should at least compare with their demos. There is a lack of depth in the discussion of how SALAD's approach differs from and improves upon previous methods except for the diffusion head, which is crucial for establishing the novelty and impact of the research.

Insufficient Result Significance:

According to Table 1, it seems that the continuous models cannot make a huge difference, which weakens the contribution of this paper. Moreover, the three variants in paper have been proposed in previous works.

**Questions:**

In Line 152, the footnote of semantic should be n instead of m since it have the same downsampling stride as the VAE.

---

### Official Review · Reviewer_wcbc · 2024-10-31

**Soundness:** 2
**Presentation:** 2
**Contribution:** 2
**Rating:** 3
**Confidence:** 4

**Summary:**

This paper proposes **SALAD**, a speech synthesis model that leverages per-token latent diffusion loss to enable continuous representations in an auto-regressive framework. Additionally, it proposes the use of semantic tokens to support variable-length modeling for speech synthesis applications. The paper includes experiments comparing discrete tokens and continuous representations on the MultiLingual Librispeech dataset to evaluate the model's performance. The results show that the proposed model achieves speech quality and speaker similarity comparable to ground-truth audio.

**Strengths:**

- The paper is generally well-written and easy to follow.
- The application of auto-regressive per-token diffusion loss to variable-length speech synthesis is novel.
- Ablations are performed for most hyperparameter choices, and the examples presented effectively demonstrate the quality of the results.

**Weaknesses:**

- Comparison to prior work is somewhat limited. The paper compares its model to XTTS-v2 but overlooks other prior work. Additionally, even with XTTS-v2, some metrics are not reported.
  - The authors point that "We did not report objective scores for XTTS due to the sample limit in their demo." While the 200-character limitation exists in the demo, the main [repository](https://github.com/coqui-ai/TTS?tab=readme-ov-file#running-a-multi-speaker-and-multi-lingual-model) has an API (tts.tts_to_file(*)) that breaks up long text into sentences for audio synthesis beyond the 200-character limit.
  - While Voicebox and Vall-E are not open-source, the ZS-TTS test setup on LS test-clean dataset and protocol are documented in detail. Voicebox, ClAM-TTs, Audiobox, NaturalSpeech3, and MELLE compare to Vall-E on ZS-TTS; is there a reason the protocol described in the Vall-E ZS-TTS protocol not followed here?
  - VoiceCraft (weights and code released), which is based on discrete codec and auto-regressive modeling would be another valuable comparison.

- The paper lacks the some ablations and misses discussion of some design choices:
  - In NAR modeling, the paper extends MaskGIT with diffusion loss which couples diffusion and MaskGIT. Ablations for (a) diffusion only and (b) MaskGIT only would be informative.
  - The paper use BPE for semantic tokens but its impact on performance is unclear. It would be make a comment on this if this was found to useful in prior work.
  - The paper describes design choices for training/inference, such as $64$ MaskGIT steps for NAR, $20$ diffusion steps, $4$ codebooks (for discrete), and $d=8$ for continuous VAE. A brief discussion of these choices would be valuable. For example, while the choice of fewer codebooks is discussed, $d=32$ appears to be more effective for continuous VAE. Similarly, Figure 6(c) indicates that $5$ diffusion steps yield better UTMOS with minor drop in the similarity scores from 0.528 to 0.520.

- Benchmarks/discussion on inference speed.
  - The combination of continuous features with a diffusion head would incur a significant inference speed cost, as each timestep now requires diffusion. Even for the NAR model, MaskGIT with a diffusion head will be slower than using either diffusion or MaskGIT alone. This is relevant for practical applications and should be discussed in the paper along with the results.
  - Section 3.3 states "Given $K$ MaskGIT steps, SoundStorm requires passes $QK$ through the transformer, as it employs a MaskGIT procedure per RVQ layer, unlike SALAD-NAR, which requires $K$ transformer passes." This should be updated to also include the effect of diffusion steps.

**References:**
- Voicebox: Text-Guided Multilingual Universal Speech Generation at Scale (NeurIPS 2023)
- CLAM-TTS: Improving Neural Codec Language Modeling for Zero-shot Text-to-speech (ICLR 2024)
- VoiceCraft: Zero-Shot Speech Editing and Text-to-Speech in the Wild (ACL 2024)
- NaturalSpeech 3: Zero-Shot Speech Synthesis with Factorized Codec and Diffusion Models (Arxiv 2023)
- Autoregressive Speech Synthesis without Vector Quantization (Arxiv 2024)
- Audiobox: Unified Audio Generation with Natural Language Prompts (Arxiv 2023)

**Questions:**

- Is speaker similarity measured with the vocoded ground truth (GT) or the raw GT? Voicebox, CLAM-TTS, NaturalSpeech3, and Audiobox report both (sim-0 and sim-R) to have a fair comparison and avoid issues related to vocoding.
- Vall-E and above papers reports both cross-sentence and continual results for ZS-TTS. Given the observed differences in speaker similarity for the AR-style Vall-E model, it would be beneficial to follow a similar protocol here.
- In addition to CER, it would be helpful to report the WER as well.

---

### Official Review · Reviewer_iCYC · 2024-11-04

**Soundness:** 2
**Presentation:** 3
**Contribution:** 1
**Rating:** 3
**Confidence:** 3

**Summary:**

The paper presents SALAD, per-token latent diffusion models for continuous speech synthesis. Unlike recent quantization-based speech synthesis methods, SALAD operates on continuous representations and is inspired by advancements in continuous modeling within the image domain. The paper extends these concepts to speech synthesis and addresses variable-length outputs, a unique challenge in audio modeling. The authors also propose discrete baselines to compare the performance of SALAD. Experimental results show SALAD achieves competitive performance in speech quality, intelligibility, and speaker similarity.

**Strengths:**

* The authors successfully apply per-token latent diffusion to continuous speech representations, which is a promising direction for speech synthesis research.
* The proposed text-to-acoustic modeling framework facilitates the parallel generation of semantic and acoustic features, effectively eliminating the need for explicit stop token prediction and potentially improving synthesis efficiency.
* The authors implements the discrete generative models to conduct quantitative and qualitative comparisons between continuous and discrete methods.

**Weaknesses:**

* The performance improvements presented are not consistently better than those achieved by discrete methods. While the paper hypothesizes that quantizing continuous latent representations is suboptimal and introduces a per-token latent diffusion approach as an alternative, the continuous VAE model (with bottleneck dimension  d=8 ) shows lower reconstruction quality, as measured by PESQ, compared to the 4-codebook Residual VQ model. Furthermore, in key generative modeling experiments, the proposed method often underperforms, as shown in Table 1 and Figure 5. I recommend that the authors clarify their rationale for choosing such a small bottleneck dimension for generative modeling. Additionally, a more in-depth investigation of such as when and why quantization of continuous latent representations leads to suboptimal performance, whether in generative modeling or reconstruction quality, would enhance the contribution and understanding of this work.
* Although the continuous model shows superior intelligibility, the paper lacks a detailed qualitative analysis or ablation study that convincingly demonstrates why continuous methods should be preferred over discrete ones. This analysis is crucial to strengthen the claims about the advantages of continuous modeling. For instance, exploring how continous methods might achieve better trade-offs between reconstruction quality of autoencoders and generation performance, or between generation performance and sampling efficiency. Demonstrating these trade-offs would offer valuable insights into the advantages of continuous representations.
* The work primarily extends existing generative methods from the image domain to audio, with the primary distinction being the parallel prediction of semantic and acoustic tokens for handling variable-length outputs. This extension, while useful, limits the scientific novelty and originality of the contribution. Emphasizing the unique challenges involved in adapting these methods to audio for variable-length modeling would be valuable. For instance, the authors could highlight how parallel semantic and acoustic token prediction is non-trivial and explain its effectiveness compared to alternative methods, such as binary stop prediction. Providing evidence that the proposed stopping condition using semantic tokens performs better than simpler binary classifiers would strengthen the originality of this work.
* The proposed method is inefficient in terms of sampling, as it requires more generation steps due to the iterative denoising process of diffusion models. Although Figure 6 (c) and (d) hint at potential advantages in sampling speed compared to discrete methods, further explanation is needed. Specifically, the authors should discuss why the generation quality degrades when the number of diffusion steps exceeds 20 and whether MaskGIT steps could be reduced while maintaining quality, possibly by applying fewer iteration steps at deeper quantization layers, as in SoundStorm. Additionally, the use of a 12-block MLP for noise estimation appears significantly larger compared to the 3-block architectures used in prior work. The authors should provide a detailed justification for using a 12-block MLP, including its impact on overall performance and whether smaller architectures were considered.

**Questions:**

* To better understand the practical impact of SALAD, including results on key efficiency metrics, such as sampling efficiency, training efficiency, or parameter efficiency, would provide a clearer view of the significance of this work.

---

### Official Review · Reviewer_qab2 · 2024-11-05

**Soundness:** 4
**Presentation:** 3
**Contribution:** 3
**Rating:** 6
**Confidence:** 2

**Summary:**

This paper studies expressive diffusion head in T2A and S2A implementations. The methods are technically sound. The results are positive. In the four claims of contributions, it is considered that the proposal of 'zero-shot speech synthesis system that uses per-token latent diffusion' is novel. Other claims are just part of the same study.

**Strengths:**

1) the idea of expressive diffusion head was studied in image generation. this paper successfully implemented the same idea in TTS;
2) the results confirmed the proposal;

**Weaknesses:**

The paper lacks a clear narrative about what authors want to achieve. It focuses on the implementation of expressive diffusion head and related TTS architecture, that makes the paper sounds a technical report than a scientific paper.

**Questions:**

1) Clarity: There are several key components in this paper, one is the expressive diffusion head, another is the variants of TTS structure. I am not clear about two things: a) what contributed to the improvement of speech quality? b) what did we expect to benefit from the expressive diffusion head?
2) Experiments: The UTMOS results for TTS is better than GT? In Table 2, I cannot clear tell whether continuous representation is better than discrete representation or otherwise. What is the take-home message of this paper?

---

### Note · Authors · 2024-11-28

**Comment:**

We sincerely appreciate the reviewers' thorough evaluations and insightful feedback. Your valuable suggestions will be incorporated to further enhance our work.

**Withdrawal Confirmation:**

I have read and agree with the venue's withdrawal policy on behalf of myself and my co-authors.